# Multistage Mechanical Activation of Multilayer Carbon Nanotubes in Creation of Electric Heaters with Self-Regulating Temperature

**DOI:** 10.3390/ma14164654

**Published:** 2021-08-18

**Authors:** Alexandr Viktorovich Shchegolkov, Sung-Hwan Jang, Aleksei Viktorovich Shchegolkov, Yuri Viktorovich Rodionov, Olga Anatolievna Glivenkova

**Affiliations:** 1Department of Technology and Methods of Nanoproducts Manufacturing, Tambov State Technical University, 392000 Tambov, Russia; Energynano@yandex.ru (A.V.S.); alexxx5000@mail.ru (A.V.S.); 2Department of Civil and Environmental Engineering, Hanyang University ERICA, Ansan 15588, Korea; 3Department of Mechanics and Engineering Graphics, Tambov State Technical University, 392000 Tambov, Russia; rodionow.u.w@rambler.ru; 4Department of Foreign Languages and Professional Communication, Tambov State Technical University, 392000 Tambov, Russia; olga-glivenkova@rambler.ru

**Keywords:** percolation, multilayer carbon nanotubes (MCNTs): multistage mechanical activation, heat release, modification, elastomers, self-regulating temperature, percolation threshold

## Abstract

The article deals with research related to the issues of nanomodification of elastomers as a basis of electric heaters with self-regulating temperature. The effect of multistage mechanical activation of multilayer carbon nanotubes (MCNTs) with graphite on the uniformity of the temperature field distribution on the surface of nanomodified organosilicon elastomer has been studied. The influence of the stages of mechanical action on the parameters of MCNTs is revealed. It has been ascertained that for the MCNTs/graphite bulk material, which has passed the stage of mechanical activation in the vortex layer apparatus, a more uniform distribution of the temperature field and an increase in temperature to 57.1 °C at the supply voltage of 100 V are typical. The distribution of the temperature field in the centrifugal paddle mixer “WF-20B” for mixing MCNTs with graphite has been investigated. It has been found that there is also a thermal effect in addition to the mechanical action on the MCNTs in the paddle mixer “WF-20B”. The thermal effect is associated with the transfer of the mechanical energy of friction of the binary mixture MCNTs/graphite on the paddle and the walls of the vessel. The multiplicity of the starting current I_p_ to the nominal I_n_ (I_p_/I_n_) is 5 for the first sample, 7.5 for the second sample, and 10 for the third sample at the supply voltage of 100 V. The effect of reducing the starting current and stabilizing the temperature indicates the presence of self-regulation, which is expressed in maintaining a certain level of temperature.

## 1. Introduction

New polymer materials are an integral part of the development of the modern energy-efficient and resource-saving industry [1]. Various approaches are used to obtain new materials. The most efficient approach is the synthesis of nanomaterials. Carbon nanostructures are of great importance among various types of nanomaterials [2,3]. The technology of obtaining nanomodified materials is one of the topical areas of research in the field of modern materials science. Nanomodification or nanostructuring opens up new horizons for the use of composite materials based on elastomers, thermoplastics, and thermosets, which will increase their resistance to both mechanical stress [4,5] and erosive wear [6,7].

Carbon nanomaterials, in particular, are also very promising as additives for creating composites with properties of protection against electromagnetic radiation in various ranges of electromagnetic radiation [8,9]. Carbon nanostructures can be effective fillers of composites for electric heaters [10,11], which have temperature self-regulation properties. It should be noted that such heaters can be used without automatic control systems [12], where carbon nanotubes are used as a nanomodifier. MCNTs are a variety of carbon nanotubes [13]. Self-regulating heaters are found to be of practical importance in the development of anti-icing systems [14]. A significant problem in obtaining nanocomposites is the uniform distribution of MCNTs in the polymer matrix. MCNTs are characterized by agglomeration and aggregation. This is due to the fact that there is an interaction between separate nanotubes (Van der Waals interaction) [15]. To eliminate the agglomeration of MCNT, methods and technological approaches associated with the pretreatment of MCNT and special technologies for introducing MCNTs into the polymer matrix are used [16,17]. The values of the percolation threshold and maximum specific conductivity are strongly correlated with the conditions for obtaining the composite, as well as with the processing methods, namely, modification, functionalization, and mechanical activation of MCNTs.

Mechanical stirring is used in most cases for the distribution of MCNTs in the polymer matrix [18]. The change in the percolation (flow) thresholds of nanocomposites in the range of 0.1 to 1.0 mass% depend on the aspect ratio of the parts of MCNTs, as well as the degree of defectiveness of MCNTs [19]. Functionalization increases the activity of MCNTs and improves the adhesion properties of nanocomposites, in particular, electrically conductive elastomers or adhesives based on them. Exohedral functionalization, associated with the formation of covalent and non-covalent chemical bonds on the surface graphene layer of MCNT, has become widespread [20]. In [20,21], a method for the functionalization of MCNTs using ultrasound treatment and oxidation with three oxidants—concentrated sulfuric acid, potassium permanganate, and hydrogen peroxide was proposed.

In the case of mechanical activation, the morphological characteristics of MCNTs can change, the bulk density of the MCNTs powder changes, the specific surface area and aspect ratio of MCNTs change.

The morphology of carbon nanostructures affects the percolation threshold of electrical conductivity, i.e., the formation of electrically conductive networks, which significantly affects the heat release when an electric current flows through the nanomodified elastomer [21].

When developing electrically conductive nanocomposites, it is necessary to take into account all the factors that can affect the formation of electrically conductive networks since conductivity is associated with at least three mechanisms: tunneling, hopping conduction, and the mechanism of activated dielectric losses [22].

It is worth taking into account the peculiarity of MCNTs synthesized by the CVD method—a wide variability of the MCNTs powder in terms of physical and mechanical characteristics while maintaining the identity of the morphological properties of separate MCNTs. This factor can affect the properties of mechanically activated MCNTs. During the synthesis of MCNTs, carbon nanotubes of different characteristics can be obtained even on the same catalyst [23], which is due to the formation of a system of agglomerates with different sizes and mechanisms of bonding between separate MCNTs within the agglomerate. The electrical conductivity of nanomodified composites is influenced by the type of conductivity of single-walled CNTs—semiconductive or metallic [24]. 

In practice, when creating electrically conductive composites, great importance is attached to the method of processing electrically conductive dispersed materials, which is associated with the need to reduce the threshold of percolation *φ_c_* and increase the value of the maximum electrical conductivity *σ_max_* [25]. One of the approaches that can affect the percolation threshold can be mechanical activation of the dispersed filler. However, the effect of mechanical activation on the percolation of polymers has been studied to a lesser extent, and in particular, this concerns MCNTs.

Mechanical activation can lead to a change in the aspect ratio, the degree of defectiveness, and fragmentation or alignment of agglomerates in MCNTs. In this case, mechanical activation will allow the creation of uncompensated chemical bonds or free radicals with a reserve of “excess” energy [25].

The influence of various types of polymer matrices on the formation of nanomodified composites should be taken into account. 

In [7,26], silicones or silanes modified graphene oxide using the Pearce–Rubinstein reaction, and as a result, it was possible to improve the dispersion in non-polar materials, such as silicone elastomers with a graphene concentration of up to 10 wt. % and improve the mechanical characteristics.

In [11,27], a decrease in the percolation threshold was observed when using a block copolymer to disperse CNTs, which indicates that the presence of a covalently bound PDMS block improves the distribution of CNTs in a silicone elastomer and allows the percolation network to form at low CNTs concentrations. Thus, it was found that the plexus of the P3DT-PDMS silicone block with bulk silicones leads to the fixation of CNTs inside the composite and provides stable conductivity during stretching and relaxation.

In [28], a combination of CNTs and nylon 6 was used, which made it possible to obtain a heater that operates at the supply voltage of 12 V. In [29], bundles of polyethylene microfibers with CuNW with silicone rubber were used to create a flexible heater. In [30], epoxy resin and CNTs were used, which made it possible to obtain a heater with the power of 2 W and heating up to 70 °C in 20 min. In [31], graphite and PVA were used, which ensured the formation of a heater with a specific power of up to 4.5 kW/m^2^ and heating up to 91 °C in 90 s. In [32,33], a complex combination of multi-directional CNTs layers filled with epoxy resin was used, which provided the formation of modes with the specific power up to 4.9 kW/m^2^ at the supply voltage of 16 V. In this case, the use of different types of polymer matrices in combination with different types of conductive fillers led to the formation of different powers of electric heating. It is preferable to use a flexible matrix—an elastomer with the ability to operate at voltages from 100 V and above, which will increase the power and reduce the operating current.

The purpose of the work is to ascertain the effect of multistage mechanical activation on the percolation of the electrical conductivity of a composite based on organosilicon elastomers and MCNT mixed with graphite, as well as to study the uniformity of the heat release during the electric current flow.

To achieve this purpose, the following tasks were conducted: -the development of a method for producing electrically conductive composites based on organosilicon elastomers modified with mechanically activated MCNTs;-the ascertainment of the effect of multistage mechanical activation on the uniformity of the distribution of MCNTs in the polymer matrix;-the study of the distribution of the temperature field in a capacitive apparatus with a high-speed rotary paddle stirrer for mixing MCNTs with graphite in a fluidized bed mode;-an investigation of the distribution of temperature fields on the surface of an electrically conductive polymer/MCNTs/graphite after mechanical activation.

## 2. Methods and Materials

### 2.1. Materials

To implement the first preparatory stage, at which there was the mixing of MCNTs Taunit-M (Nano-TechCenter, LLC, Tambov, Russia) synthesized by the CVD-method using Co-Mo/Al_2_O_3_-MgO (wt. 3%) with dispersed filler and graphite (wt. 10%) (Technografit, CJSC, Smolensk, Russia), it was proposed to use a centrifugal paddle mixer “WF-20B” (Yueyuehong, Zhejiang, China) with a rapidly rotating working body that converts bulk material into a fluidized state. Graphite has at all stages. Surfactants OP-7 (TD Sintez-Oka, LLC, St. Petersburg, Russia) the used for better distribution of MCNTs in the polymer matrix.

### 2.2. Technique of Specific Volume Resistivity Measurement 

Specific volume resistivity of heater *6* was measured using a two-wire direct current circuit using the Radiotechnika “E6-13A” teraohmmeter («Punane RET», Tallinn, Estonia) in accordance with State Standard R50499-93. The *2*-laboratory autotransformer “Resanta LATR TDGC2-1” (Resanta, Moscow, Russia) was used as a regulated power supply, at the output of which the *3* KBPC 5010 diode bridge (Solid State Devices Inc (SSDI), Firestone Blvd, La Mirada, California, USA) was installed, and the CD60 capacitor (Taizhou Huifeng Automobile Trade Co., Ltd., Zhejiang, China) was connected parallel to rectify a half-wave of the electric voltage as shown a Figure 1.

A comparative analysis of the results of measuring the specific volumetric electrical conductivity can be carried out on the basis of the equation of percolation of electrical conductivity proposed in [32]:(1)$$\sigma =\sigma_{c}+(\sigma_{m}-\sigma_{c}){\left(\frac{\phi -\phi_{c}}{F-\phi_{c}}\right)}^{t},$$
where *σ* is the specific volumetric electrical conductivity of the nanomodified elastomer, S∙cm^−1^; *σ_m_* is the specific volumetric electrical conductivity of the nanomodified elastomer at the maximum MCNTs weight content, S∙cm^−1^; *σ_c_* is the specific volumetric electrical conductivity of the composite at the percolation threshold, S∙cm^−1^; *φ* is the MCNTs volume fraction; *φ_c_* is the MCNTs volume fraction at the percolation threshold; *F* is the MCNTs packing factor; *t* is the critical indicator of electrical conductivity.

The MCNTs packing factor is:(2)$$F=\frac{m}{V\rho },$$
where *m* is mass, kg; *V* is volume, m^3^; *ρ* is MCNT density, kg/m^3^.

### 2.3. Methods for Studying the Parameters of MCNT and Heat Release of Elastomers

The specific surface area of MCNTs was determined by the BET method through nitrogen adsorption on the QuantochromeNova 1200e analyzer (Quantachrome, Boynton Beach, FL, USA). Structural studies of nanotube samples were carried out using the method of transmission and scanning electron microscopy (SEM). The samples were taken by contact with microscopic grids with adhesive composition. The investigations were carried out from different places of several samples in order to obtain better statistics on the samples under study. TEM and SEM studies were carried out using the Hitachi H-800 (Hitachi High-tech global, Tokyo, Japan) electron microscope with the accelerating voltage of up to 200 keV and Lira Tescan 3 (Tescan, Brno, Czech Republic). 

The Raman spectra of MCNTs before and after functionalization were investigated by the DXR Raman microscope (Thermo Scientific, Waltham, Massachusetts, USA). The wavelength of the exciting laser was 532 nm.

### 2.4. Technique of MCNT Mechanical Activation 

A direct-type electric drive installed on a centrifugal paddle mixer “WF-20B” provides the rotational speed of the paddles at the level of 25,000 rpm (with the supply voltage of 220 V) at the power of 3 kW and the maximum load of 1 kg, as shown in Figure 2.

Taking into account the fact that the rotation speed is not regulated, a cyclic type (stepwise) mixing mode is selected, in which there is a processing time and a pause. The total processing time can be from a few seconds to 1 h (taking into account the pauses associated with the cooling of the installation). The pause time between processing is associated with the need to cool the elements of the paddle mixer since the heat from the blades and the container is transferred to the drive elements and directly to the electric motor, which can overheat and fail. Thus, the following mode is selected: 10 min with 5-min breaks after every 20–30 s of the paddle’s rotation.

At the first stage of mechanical activation, it is possible to redistribute MCNTs in the volume and homogenize a multicomponent dispersed system, which affects the efficiency of the second main stage (Figure 3) since it is this stage that affects the activity of MCNTs when interacting with the elastomer matrix, in particular, the interfacial contact between MCNTs and the polymer matrix.

The second stage provides a reduction in the MCNT’s size and increases the efficiency of the heat release, making it uniform with a higher temperature. The first stage does not allow realizing all the possibilities of mechanical activation in full; however, at the same time, within this stage, mixing of dissimilar bulk components is possible.

At the second stage of MCNTs mechanical activation, the VLA (Vortex Layer Apparatus)-150 apparatus was used (Figure 4), the mechanical action of which is carried out due to the movement of grinding bodies (of cylindrical shape) in an alternating electromagnetic field with a frequency of up to 1 T. The time of mechanical activation in the second stage is 10 s. The electric heater is manufactured in accordance with the concept set forth in the patent [34]. A mixture of wt. 3% MCNTs with wt. 10% graphite is introduced using mechanical mixing and the addition of 1% OP-7 surfactant according to the mass of the modified elastomers.

### 2.5. Method of Preparation of Nanomodified Silicone Compound

The mechanical activation MCNTs were introduced into an organosilicon compound Silagerm 8030 (Element 14, LLC, Moscow, Russia). The Silagerm 8030 is a cold-curing organosilicon compound based on a catalytic system with Pt. After adding, the resulting composition was mixed in a mechanical stirrer for 5 min. Silicon compound was poured into a container with the volume of 100 mL (Figure 5), first component A, then MCNTs, gradually stirring. Further mixing was carried out on the mechanical stirrer “WiseStir HT 120DX” at 500–1000 rpm for 5 min. Ten grams of a hardener—component B—was added to the obtained mixture and mixed using the same stirring algorithm as in the first case [35].

#### 2.5.1. Determination of Bulk Electrical Conductivity

The bulk conductivity was measured according to the Russian National Standard (GOST) R 50499-93 method (IEC 93-80). An E6-13 teraommeter was used as a measuring device (electrical resistance range 10–10∙× 10^14^ Ohm). The method is based on two-wire resistance measurements (DC metering circuits). To measure within the other resistance ranges, a Unit 71E multimeter (Uni-Trend Technology Limited, Hong Kong, China) was employed (electrical resistance range 0.01–10 Ohm). The measurements were carried out in a special chamber, where there was a constant temperature of 20 °C and set air dryers were installed, which reduced the influence of moisture on the measurement.

#### 2.5.2. Investigation of the Temperature Field on the Heater Sample Surface 

The temperature field distribution was studied through the contactless measurement method using a Testo-875-1 thermal imager (Testo, Schwarzwald, Germany) and Fluk Ti9 (Fluk, Everett, DC, USA). The temperature was preliminarily measured with a two-channel Testo 992 thermometer (Testo, Schwarzwald, Germany). The surface temperature was determined, and based on the data obtained, it was compared with the temperature recorded by the thermal imager, whereupon the emission coefficient was selected to be used in further measurements. The resulting thermograms were processed using the IRSoft v.4.7 PC software (Testo, Schwarzwald, Germany).

## 3. Results and Discussion 

The investigation of the temperature mode of the paddle mixer, in Figure 6, showed that the paddle itself heats up to 104.6 °C and the inner walls of the container heat up to 100 °C (Figure 6a–c) due to the friction of MCNTs on the internal elements of the container under high-speed stirring. MCNTs with graphite are also heated up to 103.9 °C (histogram data, Figure 6c).

In Figure 7, SEM and TEM of the initial MCNTs and SEM graphite are shows.

Within the first stage, there are separate sub-stages, which are associated with the need to cool the high-speed rotary paddle mixer. Table 1 shows the characteristics of both the specific surface area and bulk density.

Based on the data in Table 1, the results of the specific surface area and bulk density measurements of MCNTs for the initial and mechanical activated MCNTs differ within the framework of the static measurement error at the first stage. Additionally, the second stage in the vortex layer apparatus provides an increase in the bulk density from 46.1 to 370.4 kg/m^3^, which is associated with high-energy mechanical action and intensive mechanical grinding of MCNTs. The decrease in the specific surface area is also associated with the grinding effect. In the first stage, this leads to changes in the bulk density of MCNTs from 328.1 to 334 m^2^ g^−1^. This is due to the fact that the mechanical mixing has a low energy intensive compared to the second stage of mechanical activation.

Figure 8a–c shows the Raman mapping of the surface of an organosilicon compound with MCNTs. 

The spread of MCNTs in the elastomer structure (after second stage of mechanical activation) has the form of microdimensional local agglomerate (Figure 8c), which is characteristic of nanosized carbon structures as characterized by Van-der-Vaals interaction. At the same time, for manual mixing and the paddle mixer, a strong and non-distribution is characteristic (Figure 8a,b). MCNTs mixing with graphite occurs at the first stage of mechanical activation.

When analyzing the structural features of the distribution of MCNTs/graphite in an elastomer, the percolation theory can be used [33,36]. Table 2 shows the parameters of percolation Equation (1).

Analysis of the data (Table 2) shows that elastomers E-1 and E-2 have similar values of specific volume resistance within the measurement error, while for E-3, there is a decrease in specific volume resistance by one order of magnitude. This is due to the fact that the electrical resistance of the percolation network formed by nanotubes (R_tot_) is completely determined by the resistance of nanotubes (R_t_), contact resistance (R_c_) and the number of contacts n_c_ [36]:R_tot_ = R_t_ + n_c_R_c._(3)

In case of E-3, there is a decrease in contact resistance due to the dispersion of MCNTs with the formation of a reduced resistance, which may be the result of improved wettability and a decrease in the proportion of microlocations filled with air. There is also a decrease in the effect of “stray currents”, which can be looped in agglomerates of MCNTs due to flowing through elements with lower electrical resistance.

From the data presented in Figure 9a, it follows that the sample, into which MCNTs with graphite is introduced by manual stirring, is characterized by an uneven distribution of the temperature field, with locations that heat up to 32.6 °C and locations that do not heat up, having a temperature of 23 up to 25 °C (Figure 9a). The elastomer supply voltage is 100 V.

After the first stage of mechanical activation (high-speed mixing), as shown in Figure 9b, which is mainly characterized by mixing of MCNTs with graphite, there is a redistribution of both dissimilar agglomerates and individual MCNTs and better mixing with graphite compared to the manual method. The peak temperature is 43.6 °C (Figure 9b). The unevenness of the temperature field remains. Elastomer supply voltage is 100 V. 

Figure 9c shows the distribution of the temperature field for an elastomer modified with MCNTs with graphite after the second—the main—stage of mechanical activation. The supply voltage of the elastomer is 100 V. This stage is characterized by a more uniform distribution of the temperature field and an increase in the temperature mode at the peak up to 57.1 °C. At this stage of mechanical activation, structural changes in MCNTs occur, and large agglomerates are almost completely broken (Figure 9c). Figure 10 shows spreading MCNTs in elastomer.

Figure 11 shows the dynamics of the changes in the temperature mode on the surface of the heaters.

From the data presented in Figure 11, it follows that all heaters have a self-regulation effect since there is a region with a constant temperature on the graphs, which is reached after some time (80 s). At the same time, the average heating rate is 40 °C∙min^−1^ on linear portions. Table 3 shows the time exits compared to the operating mode. 

From the data presented in Figure 12, it follows that the highest starting current corresponds to an elastomer with MCNTs and graphite at the second stage of mechanical activation, which is a consequence of a more uniformly formed electrically conductive network. The smallest value is typical of the first sample since it has separately isolated locations of the heat release, which form the level of the starting current that is in this sample. For the second sample, the starting and rated current increases since more pronounced areas of the heat release appear, which indicates the transition of the electric current into thermal energy. The multiplicity of the starting current I_p_ to the nominal I_n_ (I_p_/I_n_) is 5 for the first sample, 7.5 for the second sample, and 10 for the third sample at the supply voltage of 100 V. Thus, the starting current or activation current it is the minimum amount of charge or energy at the initial moment of the time, after this time the heating mode is exited. The effect of a decrease in the starting current and temperature stabilization indicates the presence of self-regulation, which is expressed in maintaining a certain level of temperature.

If the voltage or current level on the heater does not increase the permissible limits, then the operating time depends on structural changes (desiccation) in the polymer matrix since destruction in carbon materials MCNTs/graphite does not occur in the temperature modes, as shown in Figure 11 and Figure 12. The structural transformations in nanostructured carbon materials occur at temperatures above 400 °C in the presence of oxygen.

The conducted studies made it possible to clarify the issues associated with the effect of multi-stage mechanical activation on the properties of elastomers modified with a binary mixture of MCNTs/graphite. The obtained results make it possible to expand the understanding of the influence of the principles of mechanical activation on MCNTs, taking into account such a component as graphite and the use of high-speed mixing. In this case, one of the additional factors for improving the properties of MCNTs during mechanical activation may be a consequence of the homogeneous mixing of MCNT with such materials, such as graphite, which makes it possible to increase their electrical conductivity and leads to an increase in the specific volumetric and surface electrical conductivity of the elastomer.

It is necessary to take into account the possibility of combining the process of mechanical activation and subsequent functionalization of the surfactant, which will prevent the agglomeration in the polymer matrix.

## 4. Conclusions

A method for the production of electrically conductive composites based on organosilicon elastomers modified by mechanical activated MCNTs with the addition of graphite at the first stage of mechanical activation was developed; the authors have substantiated the multi-stage mechanical activation with two main stages: high-speed stirring (25,000 rpm) and mechanical activation in the vortex layer apparatus, as well as auxiliary stages associated with the need to cool the capacitive rotary mixer. Both the first and the second stages of mechanical activation are accompanied by the thermal action on MCNTs. The heating of the paddles and the container in which the MCNTs are located during the high-speed mixing is uneven and can cause overheating of individual elements of the mechanical activation unit.

The effect of the multi-stage mechanical activation of MCNTs on the uniformity of the temperature field’s distribution on the surface of the nanomodified organosilicon elastomer has been ascertained, and the influence of each of the stages on the parameters of MCNTs has been revealed. The second mechanical activation is characterized by a more uniform distribution of the temperature field and an increase in the temperature regime at the peak up to 57.1 °C at the supply voltage of 100 V.

The distribution of the temperature field in the centrifugal paddle mixer “WF-20B” for mixing MCNTs with graphite in a fluidized bed has been investigated, which shows that, in addition to the mechanical action on MCNTs, there is also a thermal effect associated with the transition of mechanical energy of friction of the binary mixture MCNTs/graphite on the paddle and walls of the vessel, while the temperature can reach 104.6 °C.

It was revealed that the highest starting current corresponds to an elastomer with MCNTs and graphite at the second stage of mechanical activation, which is a consequence of a more uniformly formed electrically conductive network. The smallest value is characteristic of a sample with manual stirring since it has separately isolated locations of heat release, which form the level of the starting current that is present in this sample. For the second sample (the stage of mixing in a fluidized bed), the starting and rated current increases since more pronounced regions of the heat release appear, which indicate the transition of the electric current into thermal energy. The multiplicity of the starting current I_p_ to the nominal I_n_ (I_p_/I_n_) is 5 for the first sample, 7.5 for the second sample, and 10 for the third sample at the supply voltage of 100 V. The effect of a decrease in the starting current and temperature stabilization indicates the presence of self-regulation, which is expressed by maintaining a certain level of temperature.

## Figures and Tables

**Figure 1 materials-14-04654-f001:**
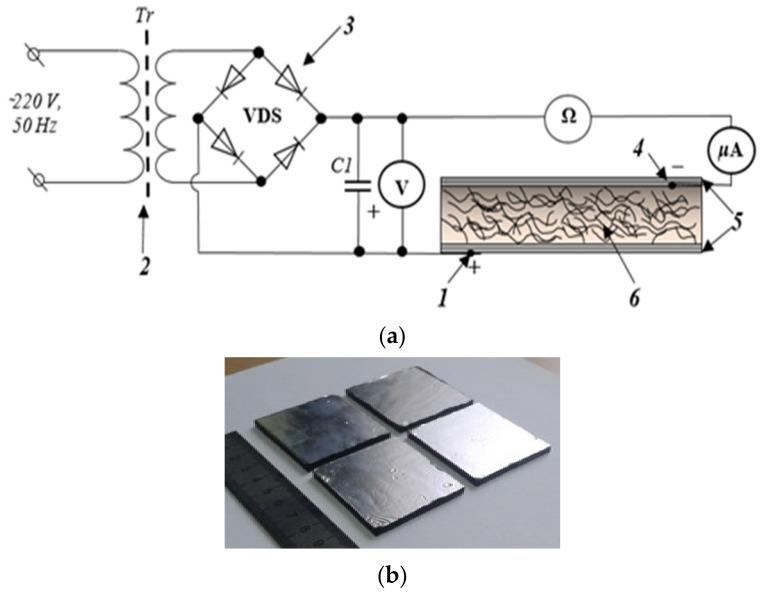
Measurements (conductivity and current) on the composites: (**a**) concept: 1—positive electrical contact; 2—LATR TDGC2-1; 3—diode bridge KBPC 5010; 4—negative electrical contact; 5—foil electrodes; 6—polymer composite Silagerm 8030/MCNTs; C1—capacitor (4700 μF); (**b**) the general view of composite heaters.

**Figure 2 materials-14-04654-f002:**
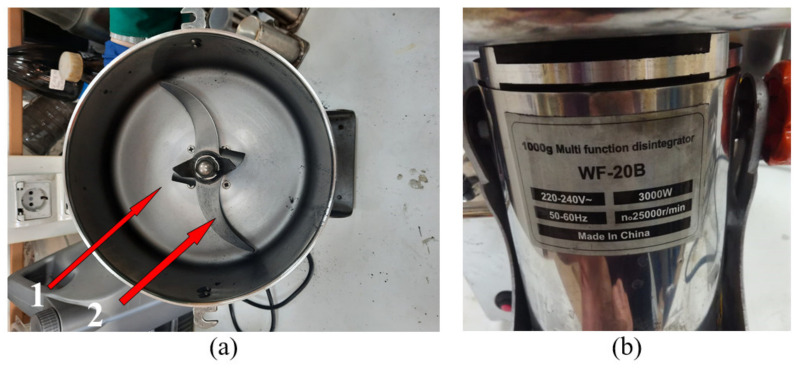
The centrifugal paddle mixer “WF-20B”: 1 is the capacity; 2 is the paddle. (**a**) view from above; (**b**) side view.

**Figure 3 materials-14-04654-f003:**
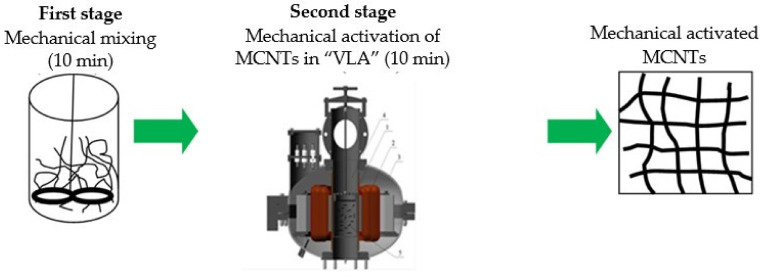
The two-stage mechanical activation of MCNTs.

**Figure 4 materials-14-04654-f004:**
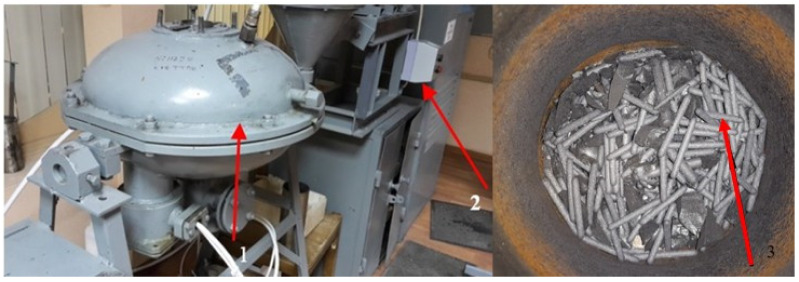
The apparatus for mechanical activation of MCNTs: 1—the working body; 2—the control unit; 3—grinding bodies (metal rods).

**Figure 5 materials-14-04654-f005:**
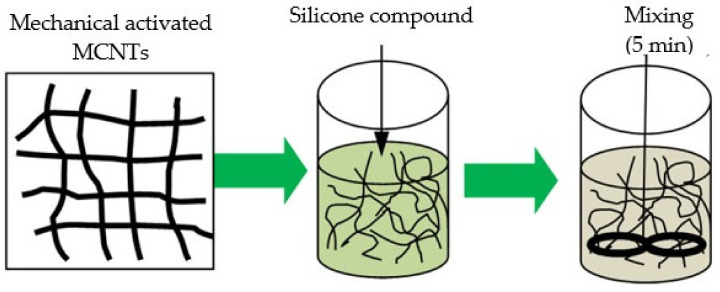
The stages of MCNTs preparation before polymer modification.

**Figure 6 materials-14-04654-f006:**
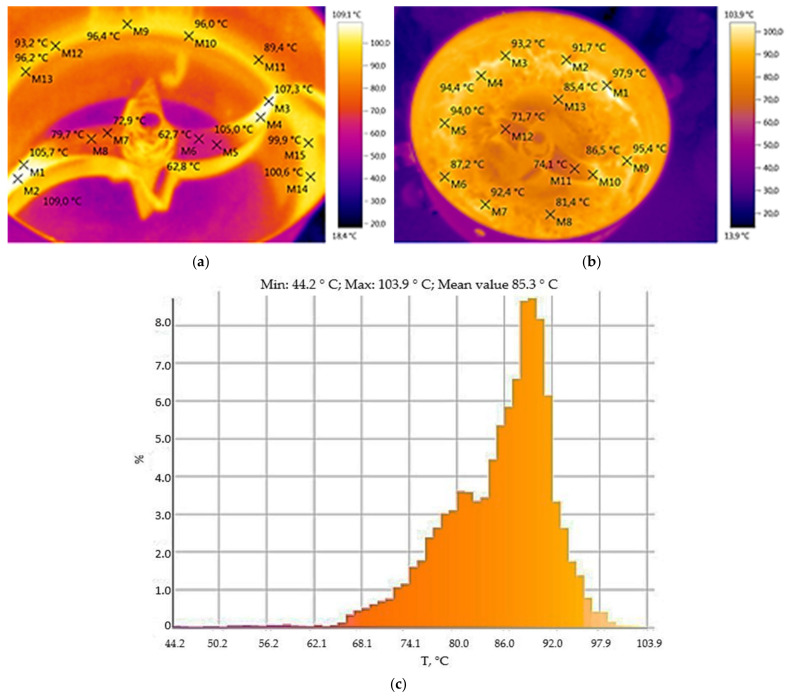
Thermal images of centrifugal paddle mixer “WF-20B” in working: (**a**)—without material; (**b**)—thermal field with material; (**c**)—a temperature histogram of the paddle mixer.

**Figure 7 materials-14-04654-f007:**
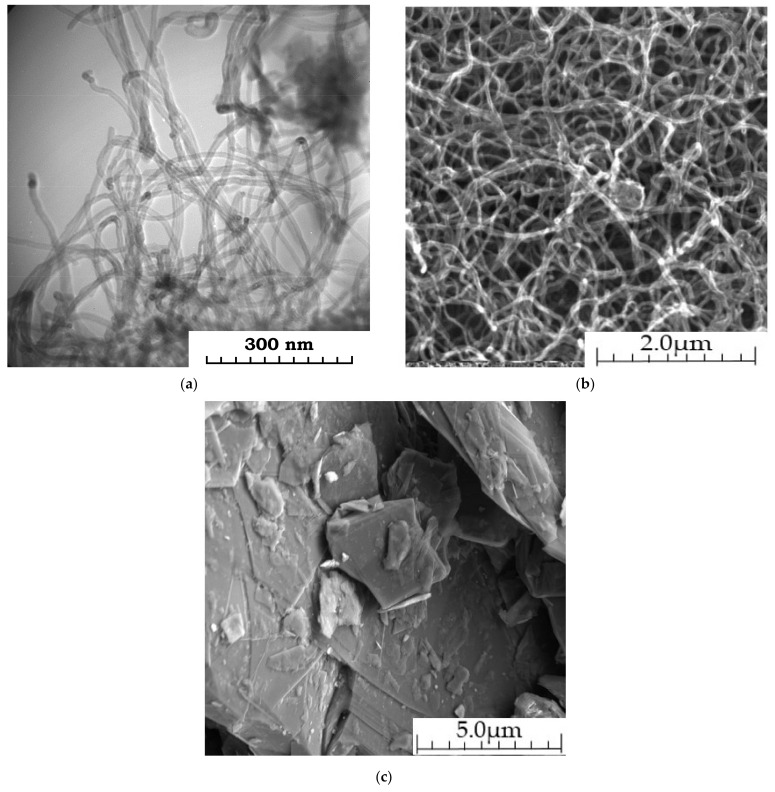
MCNTs images: (**a**) TEM; (**b**) SEM and (**c**) SEM graphite.

**Figure 8 materials-14-04654-f008:**
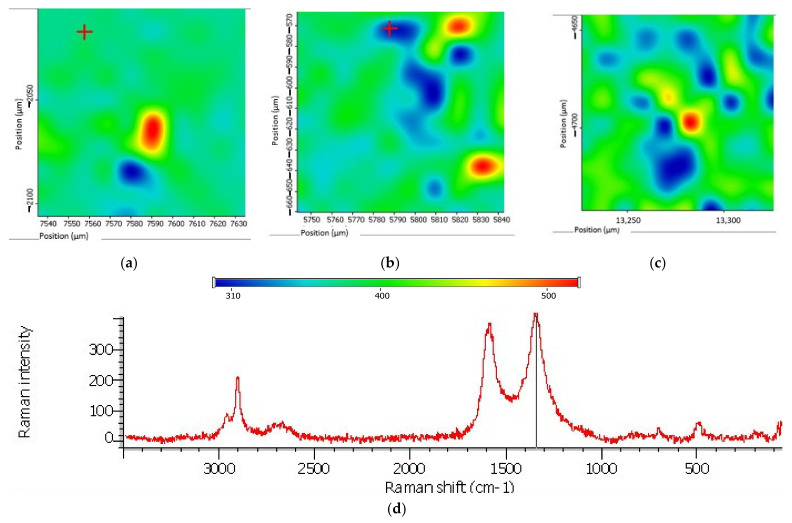
Raman mapping of the surface of an organosilicon compound with MCNTs: (**a**)—manual stirring; (**b**,**c**)—first and second stages of mechanical activation, respectively; (**d**)—Raman spectra.

**Figure 9 materials-14-04654-f009:**
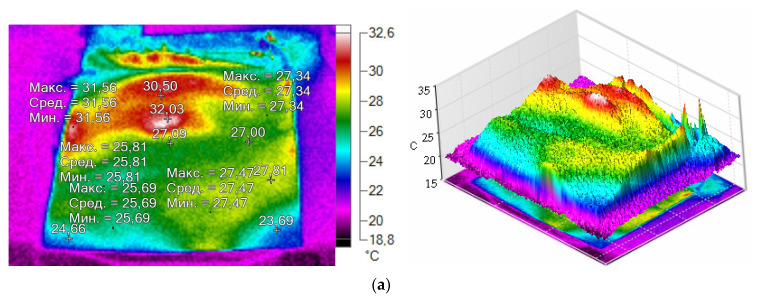
Thermal image: (**a**) elastomer with MCNT and graphite; (**b**) elastomer with MCNT and graphite: the 1st stage of mechanical activation (mixing); (**c**) elastomer with MCNT with graphite: the 2nd stage of mechanical activation.

**Figure 10 materials-14-04654-f010:**
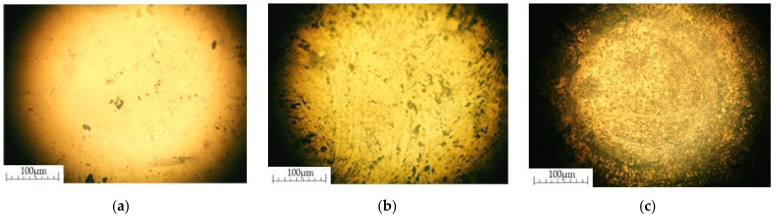
The spread MCNTs in elastomer: (**a**) composite with MCNTs/graphite; (**b**) composite MCNTs/graphite-silicone: the 1st stage mixing of MCNTs/graphite; (**c**) composite MCNTs/graphite-silicone: the 2nd stage of mechanical activation of MCNTs/graphite.

**Figure 11 materials-14-04654-f011:**
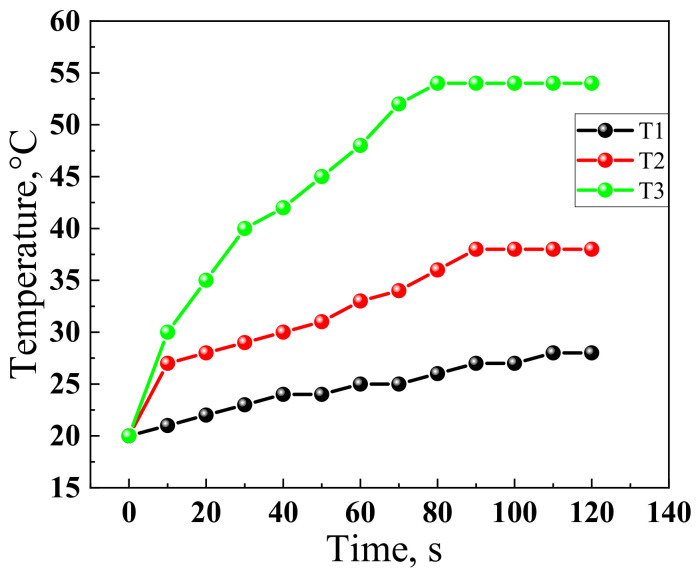
Dynamics of the temperature change on the surface of heaters: T1—elastomer with MCNTs and graphite; T2—temperature histogram for elastomer with MCNTs and graphite: the 1st stage of mechanical activation; T3—elastomer with MCNTs and graphite: the 2nd stage of mechanical activation.

**Figure 12 materials-14-04654-f012:**
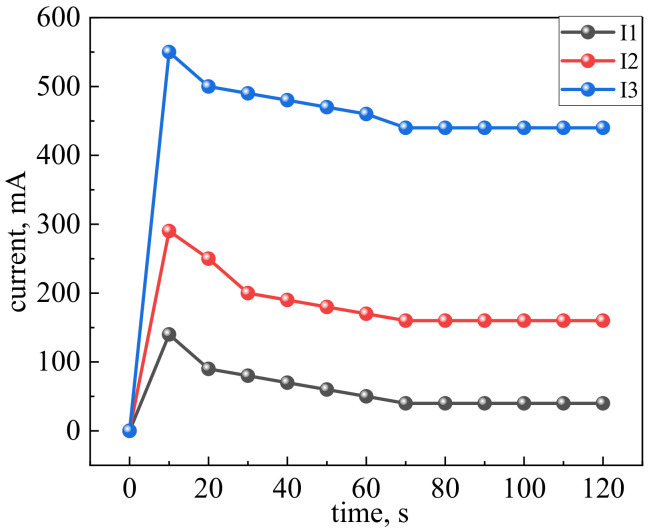
Current dynamics: I1—elastomer with MCNTs and graphite; I2—temperature histogram of elastomer with MCNTs and graphite: the 1st stage of mechanical activation; I3—elastomer with MCNTs and graphite: the 2nd stage of mechanical activation.

**Table 1 materials-14-04654-t001:** The specific surface area and bulk density of MCNTs.

№.	MCNT—Taunit M	Specific Surface Area (m^2^∙g^−1^)	Bulk Density (kg∙m^−^^3^)
1	MCNT-1	328.1	46.1
2	MCNT-2 (first stage)	334	40
3	MCNT-3 (second stage)	215.5	370.4

**Table 2 materials-14-04654-t002:** Parameters of the percolation equation for nanomodified elastomers.

Composite	σ_c_	σ_m_	φ_c_	*F*	*t*
E-1 (MCNT-1)	0.5 × 10^−3^	4.2	0.5	0.3	2.1
E-2 MCNT-2 (first stage)	0.7 × 10^−3^	5.1	0.52	0.4	2
E-3 MCNT-3 (second stage)	3.5 × 10^−2^	7.2	0.5	0.7	2.8

**Table 3 materials-14-04654-t003:** Comparison of parameters electric heaters.

Heater	Time	Area (cm^2^)	Voltage/Temperature (V)/(°C)	Refs.
Metallic glasses nanotrough	30	2.4 × 1.9	7/180	[35]
Weft-knitted carbon fabric	75	1.2 × 2.6	3.5/150	[36]
Cu–Ni micromesh	60	2.3 × 1.9	9/225	[37]
AgNW/PEDOT:PSS	107	5 × 6	6/120	[38]
Ag-grid/graphene	50	5 × 5	4/135	[39]
CNTs	180	8.5 × 3	40/50	[11,40]
E-3 MCNT-3 (second stage)	80	10 × 10	100/57.1	This work

## Data Availability

The data presented in this study are available on request from the corresponding author.

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
