# Peer review of "Multistage Mechanical Activation of Multilayer Carbon Nanotubes in Creation of Electric Heaters with Self-Regulating Temperature"

_materials, 2021, doi:10.3390/ma14164654_

Round 1

Reviewer 1 Report

Authors report a multistage method to achieve uniformly distributed carbon nanotubes within silicone network. This method involves mechanical mixing and mechanical activation. According to the data, the second stage is more critical to result in even more dispersed and distributed carbon nanotube in silicone. Thus, it achieves uniformly formed electrically conductive network. In the introduction, the authors review some prior work and clearly identify the key problem for this application: namely dispersion of carbon nanotubes. I think this paper fit Materials’ scope and standard and I recommend acceptance, after some minor revisions.

General questions:

A better characterization method that can prove E3 has more uniformed dispersion. I would like to recommend Raman spectroscopy and look at “bundle peak”

Although the introduction part is good, some important work in the field are missing.

  • Line 40: please add a citation for graphene/silicone elastomer paper: Polym. Sci., Part A: Polym. Chem. 201654, 2379–2385
  • Line 44: please add a citation: Polym. Sci. A: Polym. Chem.201553, 265-273.

Line 79 to 86: add information/background that carbon nanotube can be metallic or semi-conductive, which is critical for electrically conductive composites.

Line 212: what is the curing mechanism/chemistry of this silicone formulation? Pt or peroxide? If it is Pt, does it affect electrical conductivity?

Line 254: what is the graphite loading percentage in this composite material?

Line 267: why does contact resistance of E3 decrease? What process/formulation causes this decrease?

Reviewer 2 Report

The manuscript entitled “MULTISTAGE MECHANICAL ACTIVATION OF MULTILAYER CARBON NANOTUBES IN  CREATION OF ELECTRIC HEATERS WITH SELF-REGULATING TEMPERATURE” reports a study on the effect of multistage mechanical activation on the percolation of the electrical conductivity of a composite based on organosilicic elastomers and MCNT mixed with graphite. The uniformity of heat release during the flow of electric current was investigated.

The subject matter of the manuscript is consistent with the purpose of the journal. The introduction is sufficiently complete. However, I believe that especially the experimental part relating to the preparation of composite materials is lacking in information. Some parts are unclear. In light of the above, I believe the manuscript can be considered for publication after major revision.

Here are some tips for authors.

*) More description and information should be given in the sample preparation part. For example, no mention is given on the quantities used.

*) Little information is given on the carbon nanotubes used. For example how they were synthesized, what is the degree of purity etc.

*) Line 203. Also in this case little information is provided on the quantities used.

*) Line 212. What is the amount of nanotubes added?

*) Line 216. More information is needed regarding the user hardener.

*) What is the difference between image 5a and 5b? Perhaps image 5b is incomplete.

*) Lines 225 and 227. Is the reference figure 5 or 7?

*) The images shown in figure 6 to which stage of nanotubes are they referred?

*) Lines 226-268. I suggest reviewing the figure numbers because I believe some do not match the figures.

 *) In the version that came to me, the manuscript should be revised in its formatting. Some figures are not aligned and shifted, such as figures 3 and 4.

Reviewer 3 Report

The manuscript “Multistage mechanical activation of multilayer carbon nanotubes in creation of electric heaters with self-regulating temperature” presents the effect of materials processing on the electrical (heating) properties of MCNT/graphite composites. It is my opinion that it needs major revisions to improve its quality, readability, and interest to the reader. Additional morphological characterization would be useful in understanding the physical state of the dopants after the mechanical processes, and their dispersion in the composites. As written, the manuscript presents results, for example Figures 9-10, without meaningful discussion or comparison to other published works. The authors provide information that seems to be redundant, for example Figure 5 and Figure 8. Some of these observations are important but they would be better suited to be discussed in a Supporting Information section.

Comments:

C1: The authors should include information regarding the wt% of the MCNTs and graphite used in the composites.

C2: Line 119: The authors write “…on organosilicon and polyurethane…”. The manuscript discusses only organosilicon-based composites and not polyurethane composites. Please remove references to polyurethane.

C3: Lines 121-122. The authors write “ascertainment of the effect of multistage mechanical activation on the uniformity of the distribution of MCNT in the polymer matrix”. Morphological evaluation of the polymer composites, in support this claim, is not presented in this manuscript. Inferences regarding distribution of the MCNT in the polymer matrix are made through the heating experiments. The authors should include morphological evaluation of the composites or remove the above statement.

C4: Add a “Results and Discussion” title section. It appears to be missing.

C5: Methods and Materials section: The authors should add a detailed discussion of the electrical measurements (conductivity and current) on the composites, as well as discussion on the methodology of the composite heating experiments. Some information is already provided in the manuscript (for example heating under supply voltage of 100 V). Also, how do the authors account for environmental effects/electrical contact effects for current measurements? Surface morphology, contact resistance and environmental conditions, such as humidity, affect conductivity and current measurements.

C6: Figure 2: Fix text describing the stages of mechanical activation of MWCNs. It is hidden behind the graphics.

C7: Figure 3: Fix arrow 3 location, it does not point to the grinding elements.

C8: Figure 5 and its discussion: It is well known that mechanical mixers heat-up during operation. It is my opinion that Figure 5 and lines 223-237 discussing heating are not needed, because the heating issue was previously discussed in lines 170-177. Also the discussion of the paddle mixer heating in the abstract should be removed (lines 18-23).

C9: Table 1. The authors should further discuss the changes in bulk density.  Specifically, what is the importance in the increase of bulk density?

C10: Lines 246-248: Figure 6 and its discussion. Are these SEM/TEM images of MWCNT-1? In other words are these images of the starting material? If so, they should be presented earlier in the manuscript, before the discussion of the mixing stages. Also, what information do these images provide? From data analysis of these images, can the authors extract and report average MWCNT diameter?

C11: As written, it is not clear whether graphite is present in all stages of mixing. For example, in lines 162-163, the authors mention that MCNT are mixed with graphite. In lines 212-213, the authors write “mechanically activated MCNT were introduced…” Are MCNT-1 only multiwalled carbon nanotubes? Are MCNT-2 and MCNT-2 multiwalled carbon nanotubes or multiwalled carbon nanotubes with graphite? Please clarify. If graphite is present in all stages of mixing, what is the effect of the first stage and second stage on the graphite?

C12: Characterization of the composites: Can the authors collect optical or SEM images to show representative dopant distribution within or at the surface of the matrix?

C13: Table 2: Even though previously discussed, the authors should add units to the parameters of the percolation equation, where appropriate.

C14: Figure 8: The temperature histogram information seems redundant. The reader can deduce most of this information by looking at Figure 7. Therefore Figure 8 and its discussion should be moved to a Supporting Information type document or be completely removed.

C15: Figure 9: Is the reported temperature an average of the surface or collected at a certain location on the surface of the heaters?

C16: For all figure captions: Add “.” at the end of the figure captions.

C17: References: Use consistent formatting for all references.

Other comments:

C18: The title “Multistage mechanical activation of multilayer carbon nanotubes in creation of electric heaters with self-regulating temperature” is misleading or does not convey the message of the research. Multistage-mechanical activation is not responsible for self-regulation of the heaters; both T2 and T3 appear to self-regulate (Figure 9) and T2 has only undergone one stage of mechanical processing.

C19: Figure 9: What is the heating rate, in the nearly linear portions of these graphs? How do the heating rate/max temperature achieved compare to other published work?  

C20: Figure 10: Can the authors provide possible explanations for the observed trends in current vs time?

C21: What is the heating performance of the composites under multiple cycles of voltage application? Do re-arrangement/degradation of the electrical network occur?

Round 2

Reviewer 2 Report

I believe that the manuscript now can be considered for publication 

Reviewer 3 Report

My comments were addressed in a satisfactory manner by the authors. Therefore I recommend the revised manuscript for publication.